# Information Thermodynamics and Reducibility of Large Gene Networks

**DOI:** 10.3390/e23010063

**Published:** 2021-01-01

**Authors:** Swarnavo Sarkar, Joseph B. Hubbard, Michael Halter, Anne L. Plant

**Affiliations:** National Institute of Standards and Technology, Gaithersburg, MD 20899, USA; joseph.hubbard@nist.gov (J.B.H.); michael.halter@nist.gov (M.H.)

**Keywords:** gene regulatory networks, mutual information, channel cascades, free energy, network reducibility

## Abstract

Gene regulatory networks (GRNs) control biological processes like pluripotency, differentiation, and apoptosis. Omics methods can identify a large number of putative network components (on the order of hundreds or thousands) but it is possible that in many cases a small subset of genes control the state of GRNs. Here, we explore how the topology of the interactions between network components may indicate whether the effective state of a GRN can be represented by a small subset of genes. We use methods from information theory to model the regulatory interactions in GRNs as cascading and superposing information channels. We propose an information loss function that enables identification of the conditions by which a small set of genes can represent the state of all the other genes in the network. This information-theoretic analysis extends to a measure of free energy change due to communication within the network, which provides a new perspective on the reducibility of GRNs. Both the information loss and relative free energy depend on the density of interactions and edge communication error in a network. Therefore, this work indicates that a loss in mutual information between genes in a GRN is directly coupled to a thermodynamic cost, i.e., a reduction of relative free energy, of the system.

## 1. Introduction

Complex metabolic and regulatory functions in biology are realized through the interaction of gene products with each other. The emergent biological properties like homeostasis and differentiation are not only a function of the biochemistry of the participant genes, but also the architecture of the interactions among them [1,2]. Stuart Kauffman’s method of modeling regulatory interactions among genes as a Boolean network was established in the late 1960s [3,4]. In the last two decades, experimental characterization has provided a repository of gene network models for processes like apoptosis [5], immune response [6], embryonic development [7], and more [8].

Models of gene regulatory networks (GRN), or transcriptomic interaction networks [9], can be presented as graphs, G=(V,E), with a set of genes (or vertices or nodes), V, connected to each other with a set of edges, E. A node $v_{i}$ is connected with a directed edge to $v_{j}$, if $v_{i}$ directly regulates the expression of gene $v_{j}$. Each node is characterized by 2 degrees: the number of incoming edges to the node $v_{i}$ is the in-degree, $\deg(v_{i}^{-})$, and the number of edges emanating from the node $v_{i}$ is the out-degree, $\deg(v_{i}^{+})$. A strictly source node has $\deg(v_{i}^{-})=0$ and a strictly sink node has $\deg(v_{i}^{+})=0$. Hence, gene network models focus on the interaction between the states of the genes and coarse grain all the intermediate biochemical reactions (e.g., DNA binding, transcription, translation, etc.) that are involved in gene expression.

Graph analysis of experimentally-determined GRNs has identified attributes that are present across various species (both prokaryotic and eukaryotic) and irrespective of regulatory function, which include hierarchical organization [10], modularity [11,12] and criticality [13]. However, there is more to gene regulation than topological properties. Fundamentally, all biochemical reactions involved in gene regulation are subject to the laws of non-equilibrium thermodynamics. A thermodynamically reducible network is the one where a small subset of genes controls the free energy change that accompanies the navigation of the microstates of phenotypes. Therefore, in the context of network reducibility, it is obvious to ask what is the thermodynamic benefit of a particular gene network topology above others? Since phenotypic microstates can be represented as an energy landscape [14,15], the free energy change associated with the state of a GRN is a measure of thermodynamic benefit. To quantitatively answer the above question, in Section 2 we formulate a computational method for global transfer of information in a GRN, and in Section 3 we compute the loss of information as a field over all possible pairs of source-receiver nodes in a network. In Section 4 we use the thermodynamics of information transfer [16] to evaluate the free energy of the communication map associated with a gene network. This work establishes a method for calculating information loss in biological networks in thermodynamic terms. We use these metrics to identify the characteristics of networks that permit them to be reducible.

## 2. GRNs as Cascades of Interfering Information Channels

The topology of experimentally-determined GRNs is a topic of active research [17,18]. Topology of transcriptomic interactions across prokaryotes and eukaryotes is claimed to be scale-free [9], although a survey of biological networks has shown that the occurrence of scale-free topology is rare, but noticeably higher than other areas of application of network theory (e.g., social networks, communication networks) [19]. Therefore, we present a computational approach that is applicable to all types of GRN topologies and can identify the thermodynamic benefit of various topologies.

We use the stochastic interpretation of the model Boolean GRNs [20,21], where the state of a gene, $v_{i}$, is a Boolean random variable associated with a discrete probability distribution, $P\left(v_{i}\right)=\left\{P\left(v_{i}=0\right),P\left(v_{i}=1\right)\right\}$, with 0 as the OFF (or low expression) state and 1 (or high expression) as the ON state. Commonly, a thresholding criterion is used to map gene expression values from copy numbers to the ON/OFF states [22,23]. A directed arrow from gene $v_{i}$ to gene $v_{j}$ means either upregulation $\left(v_{i}↑v_{j}\right)$ or down regulation $\left(v_{i}↓v_{j}\right)$. Upregulation is promotion of expression of $v_{j}$ by $v_{i}$, and downregulation is repression of the expression of $v_{j}$ by $v_{i}$. The state transition equation for upregulation of $v_{j}$ by $v_{i}$ is,
(1)$$\left\{\begin{array}{c} P\left(v_{j}=0\right)\\ P\left(v_{j}=1\right) \end{array}\right\}=\left[\begin{array}{cc} 1-\rho_{0} & \rho_{1}\\ \rho_{0} & 1-\rho_{1} \end{array}\right]\left\{\begin{array}{c} P\left(v_{i}=0\right)\\ P\left(v_{i}=1\right) \end{array}\right\}$$
where $\rho_{0}$ is the probability of the input state $v_{i}=0$ erroneously producing an output state $v_{j}=1$, and $\rho_{1}$ is the probability of the input gene state $v_{i}=1$ producing an output state $v_{j}=0$. The two probability terms $\left(\rho_{0},\rho_{1}\right)$ are errors that cause a bit-flip, i.e., 1 to 0 or 0 to 1, and Equation (1) is a binary information channel model [24] for $\left(v_{i}↑v_{j}\right)$. Similarly, a binary channel model for $v_{i}↓v_{j}$ is:(2)$$\left\{\begin{array}{c} P\left(v_{j}=0\right)\\ P\left(v_{j}=1\right) \end{array}\right\}=\left[\begin{array}{cc} \rho_{0} & 1-\rho_{1}\\ 1-\rho_{0} & \rho_{1} \end{array}\right]\left\{\begin{array}{c} P\left(v_{i}=0\right)\\ P\left(v_{i}=1\right) \end{array}\right\}.$$

We will assume $\rho_{0}=\rho_{1}$ and focus on the accumulation of error due to the topology of communication. The transition matrices in the regulatory Equations (1) and (2) are the same as the matrices for information transfer through binary symmetric channels (BSC) [24]. Therefore, we can model a directed edge from an input gene to an output gene as an information channel, or more specifically a BSC. The maximum mutual information or the channel capacity of a binary symmetric channel is C(ρ)=1−H(ρ), where $H\left(\rho \right)=-\rho \log_{2}\rho -\left(1-\rho \right)\log_{2}\left(1-\rho \right)$, which is a binary entropy function. We refer to an upregulating transition matrix for a BSC with bit-flip error ρ as $T_{\mathrm{up}}\left(\rho \right)$, and a downregulating transition matrix as $T_{\mathrm{down}}\left(\rho \right)$.

Equations (1) and (2) govern the information transfer between adjacent (or nearest neighbor) genes $v_{i}$ and $v_{j}$ that are directly connected with an edge. The propagation of information between non-adjacent nodes in a GRN is subject to cumulative communication errors associated with the connecting edges and superposition, due to signaling from multiple source nodes.

The global state vector of a GRN with N nodes is a 2N dimensional vector with $P_{G}\left[2i,2i+1\right]=\left\{P\left(v_{i}=0\right),P\left(v_{i}=1\right)\right\}$. The trajectory of $P_{G}$ due to the flow of information through the network is governed by the adjacency matrix of the GRN graph. Let the adjacency matrix of the graph be A, where an element $a_{ij}$ is 1 if there is a directed edge from gene $v_{i}$ to gene $v_{j}$, or 0 otherwise. The global transition matrix for the graph, $T_{G}$, is a 2N×2N matrix. The submatrices of $T_{G}$ are defined as:(3)$$T_{G}\left[2i,2i+1;2j,2j+1\right]=\{\begin{array}{cc} \frac{1}{\deg\left(v_{i}^{-}\right)}T_{\mathrm{up}}\left(\rho \right) & \;\mathrm{if}\;a_{ij}=1\;\mathrm{and}\;v_{i}↑v_{j}\\ \frac{1}{\deg\left(v_{i}^{-}\right)}T_{\mathrm{down}}\left(\rho \right) & \;\mathrm{if}\;a_{ij}=1\;\mathrm{and}\;v_{i}↓v_{j}\\ 0_{2,2} & \;\mathrm{if}\;a_{ij}=0\\ I_{2} & \;\mathrm{if}\;i=j{\;\mathrm{and}\;\deg}^{-}\left(v_{i}\right)=0 \end{array}$$

The normalization by the in-degree, in Equation (3), assures that the effective state of a node $v_{i}$ is the superposition of all the states resulting from all the edges communicating information to the node. The last case in Equation (3) is for the source nodes in the graph and whose state remain constant during the process of information transfer [9].

Each multiplication of $T_{G}$ with $P_{G}$ updates the state of the GRN by communicating information among the nearest-neighbor nodes, which is equivalent to propagating information by one time step:(4)$$P_{G}^{\left(k+1\right)}=T_{G}P_{G}^{\left(k\right)}.$$

If the initial state of the GRN is $P_{G}^{\left(0\right)}$, then Equation (4) produces a trajectory of states $\left\{P_{G}^{\left(0\right)},P_{G}^{\left(1\right)},\cdots ,P_{G}^{\left(n\right)}\right\}$ that defines the evolution of the GRN state from the initial condition to the stationary state $P_{G}^{SS}$.

The information propagation model in Equation (4) is similar to the evolution of a multidimensional gene network probability distribution under drift and diffusion-driven Fokker-Planck dynamic. Sisan et al. [14] and Ridden et al. [15] have shown that the probability distribution from Fokker-Planck model of GRNs can be used to construct an energy landscape over the continuum gene expression state space. Our approach using information theory produces a discrete probability distribution of the GRN state, which can be used to build and discrete counterpart of the energy landscapes described in [14,15].

The state of the GRN, $P_{G}^{\left(k\right)}$, is the conditional distribution given the initial state $P_{G}^{\left(0\right)}$ after k steps of information propagation. For each step of information propagation with a time step of Δt, $P_{G}^{\left(k\right)}$ is updated by multiplication with $T_{G}$. If the initial condition of the GRN exists at $t_{0}$ then the state of a node $v_{j}$ after k steps of information propagation from source node $v_{i}$ is $P(v_{j},t_{0}+k\Delta t|v_{i},t_{0})$. This conditional probability distribution is equivalent to the solution of a Fokker-Planck model of the same GRN [25]. Hence, the thermodynamic analysis of a multidimensional probability distribution resulting from a Fokker-Planck model of GRNs is also applicable to the probability distribution $P_{G}^{\left(k\right)}$ resulting from our information propagation model.

The stationary state solution to the information propagation model $P_{G}^{SS}$ is a coarse-grained and discretized representation of the stationary state of a Fokker-Planck model of the same GRN, where values of transcription factor copy number are mapped to discrete macrostates 0 (low) and 1 (high). Therefore, the continuum energy landscape that exists for a Fokker-Planck solution to a GRN [14,15] has a discretized equivalent based on the stationary state solution $P_{G}^{SS}$ to the information propagation model.

## 3. Effective Information Loss Function for GRNs

Here, we examine how communication accuracy can affect network reducibility. How good (or lossless) is the communication from a source node $v_{i}$ to a receiver node $v_{j}$? Commonly, noise in gene expression is used to measure the loss in signal quality in genetic circuits [26,27]. The single edge communication bit-flip error, ρ, introduced in the previous section, is a coarse-grained representation of the noise in a single transcriptomic regulation step. A noiseless (or error-free) edge has a channel capacity (C) of 1 bit, and the capacity approaches 0 as ρ→0.5. So, we can quantify the loss in a single edge communication as 1−C bits. We measure the loss for any source-receiver pair in a GRN, beyond nearest neighbors, in a similar way.

Increasing loss of information due to passage through multiple edges with error ρ is expected [28]. However, the complexity of GRNs introduces other avenues for information loss: (1) Superposition of states due to information propagating from multiple source nodes, which reduces the correlation between a single source-receiver pair, and (2) the mixture of both up and downregulation edges to a receiver node, especially if these opposing signals can be induced by the same source node. We quantify the loss for a source-receiver pair under the conditions that causes maximum interference from the other nodes.

The highest entropy state of a node is $P_{max}=\left\{0.5,0.5\right\}$, which is also the input state at which a BSC achieves the channel capacity [24]. If we set the state of all the source nodes in the GRN to $P_{max}$, then at the stationary state of the GRN, $P_{G}^{SS}$, the state of the all the nodes in the GRN is also $P_{max}$. If we change the state of a source node $v_{i}$ to {1,0} and find that a receiver node $v_{j}$ is still at {0.5,0.5}, then there is high loss of information from $v_{i}\to v_{j}$. On the other hand, if the relative entropy of the state of $v_{j}$ is low with respect to the state {1,0}, then the information loss is lower.

The actual steps for quantifying the loss function from source node $v_{i}$ to a receiver node $v_{j}$ are the following:
(1)Compute the stationary state solution to the GRN for two initial conditions: (a) $P_{i,OFF}\equiv P_{G}^{\left(0\right)}\left[2i,2i+1\right]=\left\{1,0\right\}$, and (b) $P_{i,ON}\equiv P_{G}^{\left(0\right)}\left[2i,2i+1\right]=\left\{0,1\right\}$, with the rest of the source nodes at $P_{max}$. The solution at a receiver node $v_{j}$ is $P_{G}^{SS}(v_{j}|P_{i,OFF})$ and $P_{G}^{SS}(v_{j}|P_{i,ON})$, respectively.(2)Construct the effective transition matrix for communication from $v_{i}\to v_{j}$ as:(5)$$T_{\mathrm{eff}}\left(i\to j\right):=\left[\begin{array}{cc} P_{G}^{SS}\left(v_{j}|P_{i,OFF}\right) & P_{G}^{SS}\left(v_{j}|P_{i,ON}\right) \end{array}\right]=\left[\begin{array}{cc} P_{G}^{SS}\left(v_{j}=0|v_{i}=0\right) & P_{G}^{SS}\left(v_{j}=0|v_{i}=1\right)\\ P_{G}^{SS}\left(v_{j}=1|v_{i}=0\right) & P_{G}^{SS}\left(v_{j}=1|v_{i}=1\right) \end{array}\right].$$(3)Compute the loss function in bits for communication from $v_{i}\to v_{j}$ as:(6)$$L\left(i\to j\right)=1-c\left(T_{\mathrm{eff}}\left(i\to j\right)\right).$$

The second term in Equation (6) is the channel capacity in bits for the effective transition matrix. The loss function defined in Equation (6) is a field over all existing pairs of source-receiver combinations in a GRN. By definition, L(i→i)=0, and L(i→j)=1 if there is no path from $v_{i}\to v_{j}$.

We demonstrate the loss function, Equation (6), using numerical results from model graphs generated using the Barabási–Albert preferential attachment model (Appendix A) [29]. All of our analysis uses graphs with 100 nodes. Two parameters are used to control the graph generation process: (1) The in-degree of every node in the graph, m, while placing no constraint on the out-degree, and (2) the ratio of downregulation edges to upregulation edges in the graph, β (Appendix A). The in-degree to a node is the number of other nodes that can directly regulate that gene. Hence, in our simulation we have assumed that every gene in the network is directly regulated by m other genes. Obviously, the in-degree is inhomogeneous in a real GRN, but this assumption allows us to conveniently study the impact of increasing density of direct transcriptomic regulation in a GRN on the global information loss. Our method of information propagation and subsequent analysis is not restricted to the model GRNs chosen for demonstration and is applicable to all types of directed graphs.

Increasing m increases the number of nodes in the GRN that have a path to a single node, which we refer to as the accessibility score (Appendix A). This is illustrated in Figure 1a using three Barabási–Albert graphs with m=1, 2, and 3, respectively. Every node is shaded in proportion to the number of other nodes in the graph that can access it—a node with a darker shade means more nodes have a path to it. Rather than the distribution of shades in a single graph in Figure 1a, it is more important to note the global prevalence of darker shade nodes with increasing m. The increasing fraction of darker shaded nodes means an increase in global accessibility across all the nodes in the network (Appendix A). The mean accessibility score, or the average accessibility to a node from all other nodes in the network, increases with m by design.

The other factor that can reduce the effective information transfer is the mixture of up and down regulation signals to a given node in the network. Figure 1b shows that how increasing the ratio of downregulation edges to the upregulation edges in the graph, β, increases the number of nodes in the graph that are receiving mixed signals, $n_{mixed}$. If the signal from a source node forks into two separate pathways to a receiver node, and one path ends with an upregulation edge and the other with a downregulation edge, then the effective information transfer to the receiver node is reduced.

As illustrated in Figure 2a, the state of a receiver node, $v_{j}$, is determined by the states of all contributing source nodes, using Equations (3) and (4). The 3rd panel of Figure 2a shows that when all the source nodes are at maximum entropy, the receiver node is also at maximum entropy and independent of up or downregulation and the edge bit-flip error, ρ. On the other hand, when a single source node, $v_{i}$, is at low entropy, then the bit-flip error values for the edges on the source-receiver path determine the state of the receiver node as shown in the first and second panels of Figure 2a. Furthermore, the state of the receiver node is superposed with the maximum entropy state of the other source nodes. Therefore, the low entropy input from a single source gains entropy as a function of the edge bit-flip errors and from superposition from other sources. The information loss field computation using Equations (3)–(6) determines the effect demonstrated in Figure 2a for GRNs involving a large number of genes and complex information propagation pathways.

When we evaluate the loss field for every source-receiver pair in the model GRNs shown in Figure 1a, we notice that the information loss due to superposition increases markedly with increasing m, as shown in Figure 2b. The sensitivity of the loss field to the in-degree m, also depends on the edge bit-flip error value ρ. When the bit-flip error is small, ρ=0.01 (1st row in Figure 2b), then the contrast between the loss field for m=1 and m=3 is significant, increasing approximately from 0.2 bits to 0.9 bits. When the bit-flip error is larger, ρ=0.1 (2nd row in Figure 2b), then the increase in loss field from the m=1 type GRN to m=3 is smaller, approximately from 0.8 bits to 1 bit. Hence, the loss field quantifies the effective deterioration of signaling due to combination of superposition and edge communication errors. Though the m=3 type GRN has more source-receiver pairs compared to the m=1 type GRN, abundance of accessibility reduces the quality of communication as apparent in the respective loss fields.

As evident from the loss fields in Figure 2b, a low entropy input of $P_{i,OFF}$ or $P_{i,ON}$ from a single source node can be diminished if high entropy information from the rest of the source nodes in the graph is superposed on the receiver, leading to a high global entropy for the network. Therefore, for graphs with a high mean accessibility score, which increases with m, it is harder to control or correlate the state of all the nodes in the GRN using a single source node without cooperation from other source nodes. The increase in information loss with increasing m is most prominent for the dominant source nodes, which can send information to all the nodes in the graph (near 0 on the source node axis in Figure 2b).

Increasing the ratio of up and down regulation edges (β) for a fixed GRN increases the loss field value only for the dominant source nodes as shown in Figure 2c, which in this example are the first five source nodes (i<5). Increasing the mixture of up and down regulation does not change the loss field for the lower ranked source nodes, i.e., the source nodes that can propagate information to only a small subset of the receiver nodes in the GRN. Moreover, comparing Figure 2b,c reveals that information loss is more greatly affected by the increase in superposing pathways (i.e., m) than by the increasing mixture of up and downregulation.

The large difference in loss field contrasts between m=1 and m=3 in Figure 2b suggests that we can claim that network of type m=1 allows for an ideal master regulator that can communicate to all the other nodes in the GRN with minimal information loss when the communication error in every single edge is low. The value of loss for the m=3 GRN is high because of the existence of many pathways, so it is challenging for a single node (or gene) to emerge as a master regulator. Therefore, a relatively low number of superposing pathways supports the existence of a master regulator and can be an indicator of a reducible network, unless the communication error in the edges is very high.

## 4. Relative Free Energy and Reducibility of GRNs

The method of calculating the effective transition matrix, Equation (5), and the loss field, Equation (6), has a direct thermodynamic interpretation. Low information loss between a pair of genes means the network topology and the edge communication error values are such that there exists high mutual information, or correlation, between the states of two genes. Parrondo et al. has shown that the existence of high mutual information between the two components of a system equates to a proportionate increase in the nonequilibrium free energy of the system [16]. Since the amount information loss, or mutual information, is a consequence of the information propagation in GRNs, Equation (4), we can effectively compute the free energy change associated with the information propagation.

More specifically, a lower information loss, Equation (6), from a source gene $v_{i}$ to a receiver gene $v_{j}$ means when the source node is at low entropy then the receiver node is also close to a low entropy state. But if the information loss from $v_{i}$ to $v_{j}$ is high, then the receiver node is closer to the maximum entropy state. A set of low loss values from a single source node to all the other nodes in the network, like for the source node $v_{0}$ in the m=1 type GRN for ρ=0.01 shown in Figure 2b, means a single source node shifts all the other nodes in the network close to a low entropy state. The relative entropy of the state of an individual node with respect to the maximum entropy state, $P_{max}$, provides the relative free energy of a single node. Summing over this relative entropy over all the nodes in the network determines the relative free energy induced by the single low entropy source node. Therefore, the reduction in entropy of all the nodes due to information propagation results in an increase in the free energy of the network with respect to the maximum entropy state of the network.

The highest entropy state of a network is the equilibrium state where each node is in the maximum entropy state, $P_{max}$. Changing the state of a single source node, either to $P_{i,OFF}$ or to $P_{i,ON}$ and propagating the information using Equation (4) to achieve the stationary state, $P_{G}^{SS}$, results in moving individual nodes from the highest entropy state to a lower entropy state. The relative free energy associated with the global lower entropy stationary state $P_{G}^{SS}$ is,
(7)$$\frac{1}{k_{B}T}\Delta F\left(P_{i,ON}\right)=\underset{j\in V}{\sum }\underset{a\in \left\{0,1\right\}}{\sum }P_{G}^{SS}\left(v_{j}=a|v_{i}=1\right)\log_{2}\frac{P_{G}^{SS}\left(v_{j}=a|v_{i}=1\right)}{P_{max}\left(v_{j}=a\right)}$$
where $P_{G}^{SS}(v_{j}|v_{i}=1)$ is the stationary state of node $v_{j}$ when the source node $v_{i}$ is ON. We can similarly compute a free energy change due to $P_{i,OFF}$ or due to any other state of the input, $P\left(v_{i}\right)$. Since each edge in the model GRNs is a binary symmetric channel, the free energy change in the network due to setting a node $v_{i}$ to $P_{i,ON}$ or $P_{i,OFF}$ is the same.

Therefore, we anticipate that the lower loss field for source node $v_{0}$ for the graph m=1 shown in Figure 2b means that a single source node can push the entire GRN to a lower entropy more successfully than the other two cases (for m=2  or 3). So, for m=1  type graphs the relative free energy of the GRN due to the low entropy state of source node $v_{0}$ should be higher than for graphs where m>1.

In Figure 3 we present network relative free energy distributions resulting from edge errors, as a function of signal superposition. Unlike the loss field results in Figure 2b,c, which were for graphs with the same communication error value ρ, we assumed that the communication error for an edge is a uniformly distributed random variable in the domain [0,0.5]. The distributions in relative free energy for each type of network, i.e., *m* = 1, 2, or 3, were obtained by simulating 5000 replicates of a graph with the same connectivity but a different set of error values for the edges, sampled from the uniform distribution U[0,0.5]. An example of type m=1 network with a random edge communication error field is shown in Figure 3a. This calculation is similar to observing the relative free energy distribution in a cell population, where each cell has the same GRN topology but there exists a variability in edge communication errors within each cell’s network. If the distribution in the edge errors, ρ, is narrower than a uniform distribution the result will be a reduced variance in the relative free energy distributions shown in Figure 3.

The relative free energy distribution for m=1 (Figure 3b) is asymmetric, but for GRNs with high number of superposing pathways, as in m=3 type graphs, the relative free energy is distributed like a normal distribution. The broader distribution suggests that the relative free energy of each replicate network simulation is uncorrelated due to increasing interference. Correlation among replicates is a combined consequence of m and the edge communication error values. If the edge errors are distributed in low range of values, e.g., uniformly distributed between [0.0,0.1] then in spite of the effects of superposition, the probabilistic states (the global state vector $P_{G}^{SS}$) of the replicates will be closer to each other. However, when the edge errors vary over a wider range, e.g., uniformly distributed between [0.0,0.5], then increasing m, which increases the number of edges and pathways for information transfer, increases the variability in the probabilistic GRN states among replicates. Hence, if a GRN has a high mean accessibility score, then the relative free energy values present in individual cells in a population are more uncorrelated with each other. Since experimentally observed phenotypic manifestations caused by a GRN are a function of the free energy change that are induced by a GRN [14,15], we claim the distributions in observed phenotypes are analogous to the distributions in ΔF, especially for the graphs with lower mean accessibility score.

Performing the relative free energy calculations for multiple source nodes in the model GRNs, instead of only the most dominant one, reveals a thermodynamic criterion for reducibility. Figure 4a shows the relative free energy distributions for the top ten source nodes (ranked by the number of other nodes they can send signal to) in the model GRNs due to a uniformly distributed edge error value. An order exists in the relative free energy distributions as a function of source nodes for m=1 type graph. Not only does the source node $v_{0}$ induce significantly higher relative free energy compared to the other source nodes, but also the median value of ΔF for the m=1 graph is higher than the value for m=2 and m=3 graphs. Therefore, the relative free energy distributions are a criterion for thermodynamic hierarchy for source nodes and help to identify candidate master regulators in GRNs. Comparison of the ΔF distributions for multiple source nodes reveals whether that hierarchy exists or not. We claim that the existence of a strongly resolvable hierarchy, i.e., ordered median ΔF values and low overlap in the ΔF distributions for different source nodes, implies that the GRN is thermodynamically reducible. In a network with a small m value, most of the communication to other genes originate from the source node that has the highest out-degree, which creates an outgoing communication hub. Whereas, in a network with a large m value, there are multiple pathways for communication among genes in addition to the ones originating from the outgoing hub. However, the presence of several communication pathways is accompanied with the cost of a lower inducible relative free energy and the lack of hierarchy among the source nodes (Figure 4a). Interestingly, outgoing hubs have been observed in naturally-occurring GRNs [30,31], which may be justified using the thermodynamic hierarchy resulting from the relative free energy distributions.

The existence of the order in ΔF distributions is a function not only of topology and also of the distribution in the edge communication error values. We demonstrate this in Figure 4b using the ΔF distributions for m=2 type graphs, but with increasing the range of values of ρ. When the edge error value is uniformly distributed within a more constricted range, ρ∈[0,0.1], we still observe a strong hierarchy in ΔF distributions—the median ΔF values for different source nodes are separated beyond the dispersion in the individual distributions. However, this hierarchy is lost upon increasing the extent of variability in ρ to uniformly distributed in [0.0,0.5], the ΔF distributions for different source nodes become similar to each other, and the median ΔF values decrease for all the source nodes compared to the two narrower distributions in ρ. Thus, increasing variability in edge error values diminishes the possibility of the existence of a small subset of thermodynamic master regulators.

The choice of a probabilistic edge error field instead of a fixed error value for all edges is a better model for real biological GRNs. For a specific regulatory process, the set of intracellular reactions is the same for all cells in a steady state population. We explicitly considered variability in ρ, which could result from stochastic fluctuations in concentrations, binding rates, diffusion, etc, due to heterogeneity in the internal environment of the cells. Therefore, the variability in the edge error values result in the distributions of ΔF. In fact, experimental observations of the heterogeneity in gene expression in steady state distributions of cell population phenotypes resulting from [14] are highly reminiscent of the frequency distributions shown in Figure 3. We have previously demonstrated that distributions of phenotypes in cell populations represent microstates of a potential landscape, which is consistent with these observations of distributions in ΔF.

## 5. Conclusions

Scale-free or power law topologies are popular models for biological regulatory networks. We found that even within these topological classes, the quality of information transfer can vary due to interference of signal from multiple sources and superposition of up and down regulation signals. We developed the concept of a loss field to quantify the pairwise communication among nodes, and the algorithm to compute this loss field. The loss field can be used to identify potential master regulators by determining the quality and uniformity of communication from a single node to all the other nodes in the network. Relatively low connectivity is necessary for the existence of a master regulator and is an indicator of a reducible network. In the absence of high edge errors, a source node in a network that has fewer superposing pathways is more influential for communication efficiency and that network is more likely to be reducible.

We found a fundamental connection between the magnitude of information loss and the relative free energy that can be induced in a network using a single source node, i.e., without co-operation (or correlation) with other source nodes. Moreover, the relative free energy distributions induced by individual nodes emerge as a criterion for a thermodynamic hierarchy of source nodes (and identification of candidate master regulators) in GRNs. We claim that the existence of a strongly resolvable hierarchy, i.e., ordered median ΔF values and low overlap in the ΔF distributions for different source nodes, means the GRN is thermodynamically reducible. Calculation of this free energy for a variable communication error field produces distributions of the inducible free energy change that serve as a signature of the quality of communication. Specifically, if the information loss is high then the distribution in relative free energy of the microstates of the network is closer to a normal distribution. On the other hand, if the information loss is low, and there is a dominant node, then this inducible relative free energy distribution is asymmetrical. Therefore, the deviation of the relative free energy distribution from a normal distribution is associated with lower information loss, higher relative free energy, and a more reducible network. By calculating the relative free energy change that can be obtained by different nodes in a network, ranked according to their accessibility to other nodes, we can determine how many nodes are required to achieve a threshold relative free energy. Hence, our combined approach of information propagation followed by relative free energy calculation informs us about the minimum set of nodes in the network that are relevant to determine the thermodynamic states of the network.

## Figures and Tables

**Figure 1 entropy-23-00063-f001:**
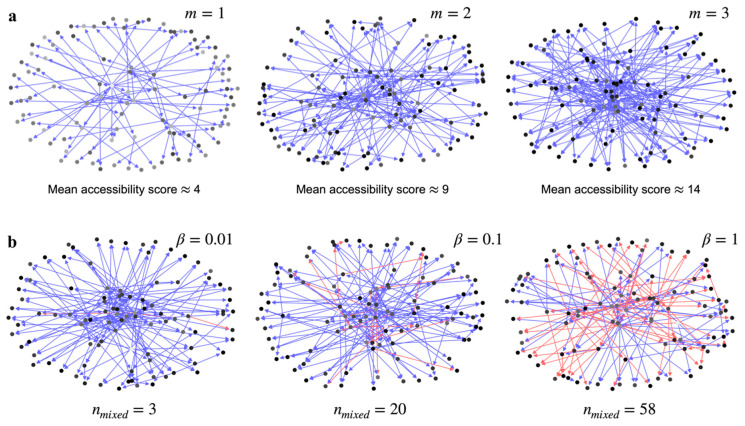
Topological factors that increase the information loss field, Equation (6). Blue arrows represent upregulation edges and red arrows represent downregulating edges. (**a**) Model GRNs with 100 nodes generated using the Barabási-Albert model. Each of the graphs has a fixed in-degree for every single node (m). There are no constraints on the out-degree. Higher values of m and the mean accessibility scores of the graphs indicate greater global connectivity between nodes in the graph. More highly accessible nodes are indicated by a darker color. A high accessibility score increases signal interference and reduces the effective channel capacity between a single source-receiver pair. (**b**) The effect of a mixture of up and down regulation edges between nodes for graphs of type m=2. β represents the ratio of down-regulating edges to up-regulating edges in the graph. $n_{mixed}$ is the number of nodes in the graph that are receiving both up and down regulating signal. Increasing β increases the number of nodes that can receive mixed signals.

**Figure 2 entropy-23-00063-f002:**
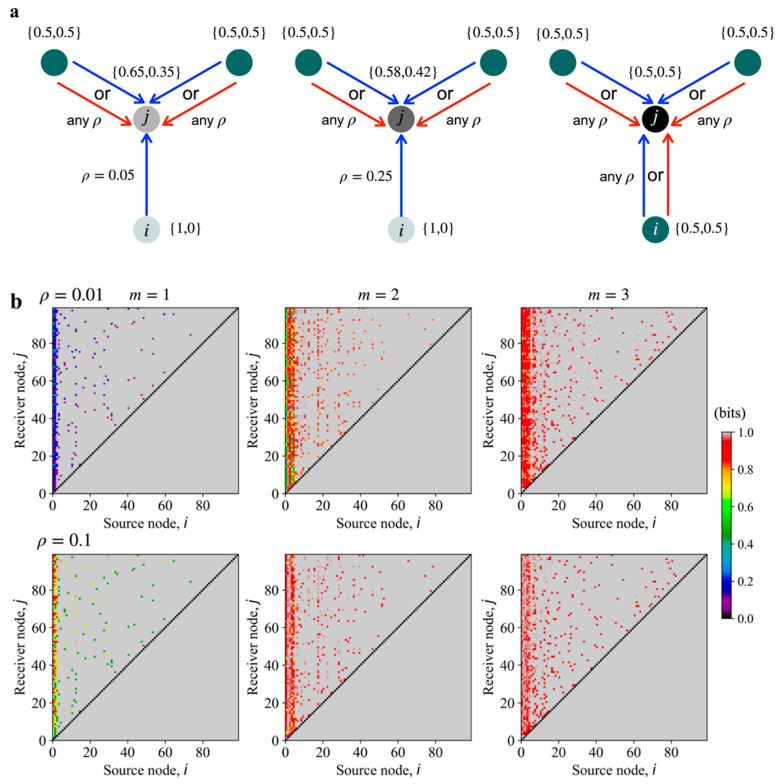
Information superposition and loss field, as defined in Equation (6), for the model GRNs shown in Figure 1. (**a**) Effective state of entropy (indicated by color opacity, higher opacity corresponds to higher entropy) at a receiver node (colored black) due to edge communication errors and interference from other source nodes (colored green). Blue arrows represent upregulation edges and red arrows represent downregulation edges. The numbers in the braces represent probabilistic state of the individual nodes as {P(v=0),P(v=1)}. The exact error values are important only for the edges that are in the source-receiver path. Maximum entropy inputs at all source nodes results in the maximum entropy state at the receiver node independent of the edge error or the type of regulation. (**b**) Loss field values for model graphs showing the effect of increasing superposition, as a function of increasing m with β=0 and with two different values of edge communication error ρ. The first row of loss fields is for ρ=0.01 and the second row is for ρ=0.1. The nodes are numbered in the descending order of their access to other nodes, i.e., node 0 can send signal to most of the other nodes in the graph and node 99 does not send information to any other node. The loss values are the lowest for the source node 0, which is the node with access to most of the other nodes in the graph. If a receiver node $v_{j}$ is inaccessible from source node $v_{i}$, then L(i→j)=1 bit by default. (**c**) Loss field values for GRNs with mixture of up and downregulation, as shown in Figure 1b. ρ=0.01 for these loss fields. Increasing the ratio of down regulation to up regulation increases the loss only for the dominant source nodes (i≤5 in this example). For (**b**) and (**c**), the color bar scale indicates the loss field values, L(i→j) as determined using in Equation (6).

**Figure 3 entropy-23-00063-f003:**
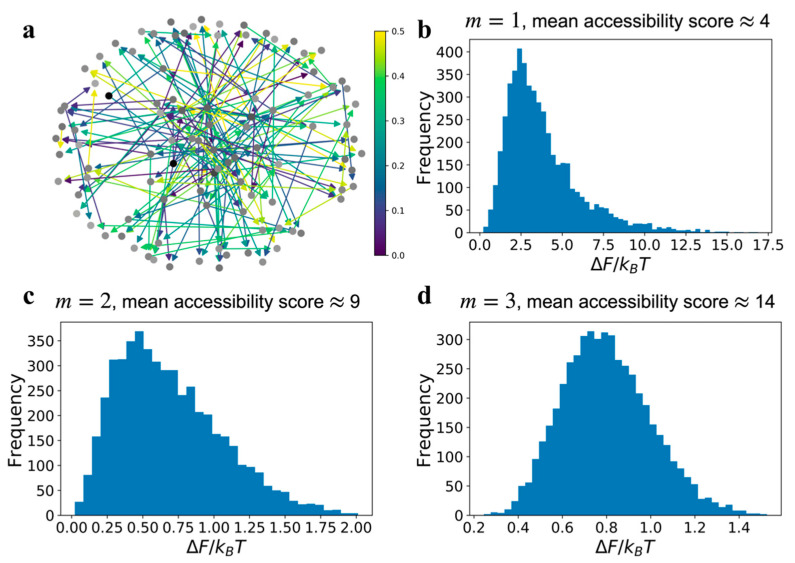
Distribution of relative free energy of networks with different mean accessibility scores and a uniformly distributed edge error field, ρ∈U[0,0.5]. (**a**) The graph shown is one among the 5000 realizations of a random error field on a type m=2 Barabási-Albert graph, which has a mean accessibility score ≈9. (**b**–**d**), show the resulting distributions in relative free energy over 5000 realizations due to the same uniformly distributed error field, but for graphs with different accessibility scores originating from the choice of m.

**Figure 4 entropy-23-00063-f004:**
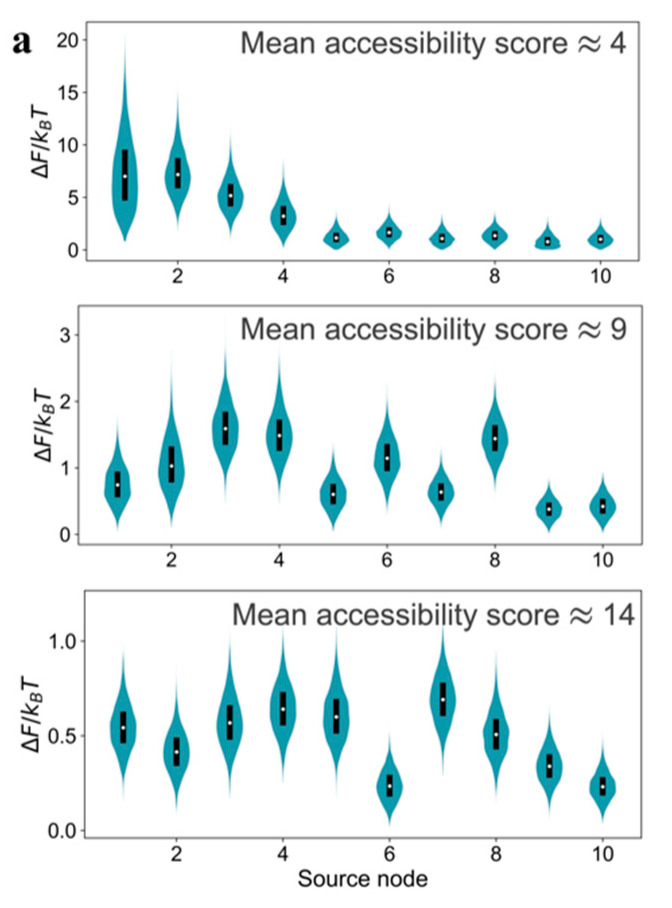
Ordering in the inducible relative free energy distributions caused by a variable edge communication error field. (**a**) Relative free energy distribution of the top ten source nodes for m = 1,2, and 3 type graphs, which have mean accessibility scores 4, 9, and 14, respectively. The communication error for every edge in the graphs were assumed to be uniformly distributed in the domain [0.0,0.5]. (**b**) Free energy distribution for top ten source nodes in type m=2 (mean accessibility score 9) for increasing domain of variability in the edge communication error value.

## Data Availability

The software used to generate the model GRNs and produce the results in this study are available on request from the corresponding author.

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
