# Peer review of "Information Thermodynamics and Reducibility of Large Gene Networks"

_entropy, 2021, doi:10.3390/e23010063_

Round 1

Reviewer 1 Report

Referees Comments on previous manuscript.

The authors have used methods in graph analysis to demonstrate that information loss in genetic regulatory networks can be inexplicitly modelled. In this manuscript, the authors have worked out their concept in great detail and I appreciate their attempt to make a connection to the thermodynamics laws that govern biochemical networks.

  1. Connection information loss and thermodynamics. The extend to which information loss and what specific aspect in thermodynamics are related, however, require clarification. The authors stated in the abstract (line 21): “Therefore, this work connects information loss in biological networks to thermodynamics”. Could the authors specify what they mean with this statement; (I) does ‘connects’ means that one influences the other? The abstract, would benefit from an additional sentence that discusses the consequence of the connection (a summary of what the authors conclude in lines 346-350). (II) Do they consider all biochemical networks to behave in similar ways as the genetic regulatory networks, under the assumed conditions in this manuscript?
  2. Thermodynamic ‘benefit’. Could the authors specify what is meant by a ‘thermodynamic benefit’ (Line 47)? (I) Are there assumptions for the system in which these benefits are considered: that is, does this manuscript consider both thermodynamically open as well as closed systems? (II) Related to these questions, a classical paper that deals with the relation between information theory and genetic regulatory networks is “Eigen, M. (1971) Naturwissenschaften, 58, 465-523.” This paper is not cited but may be useful for this manuscript. In here, Eigen states that information loss provides a pathway to selective advantage in genetic regulatory networks but that a balance between the rate of production of information, and the quality of information is required. Could the authors comment on the omission of rates of information production in their hypothesis to define a thermodynamic benefit?
  3. From a GRN to a Boolean GRN. The authors hypothesize an On and Off state (section starting from line 62), which builds on cited references 18 and 19. GRNs are considered to comprise all intermediate biochemical reactions (line 40) but the assumptions that went into their model to consider only high (defined as ‘On’) and low expression levels (defined as ‘Off’) is not explained explicitly. I believe this section requires a greater level of detail to help the reader understand how to translate a GRN into a Boolean GRN.
  4. Hidden validations. The authors refer to a seemingly important assumption from reference 20 to build there model in lines 72 and 129. The story in this manuscript would benefit from a clarification why “Thus, each directed edge from one gene to another is equivalent to a communication channel (line 75)”. Similarly, the authors refer to references 21 and 22 to claim the observed relationship between distributions of phenotypes and distribution of Delta F (see main text Fig. 3). The distinction of prior art and their claim would be clearer, than it is currently written in lines 278-280, if they could use the a structure in the form of: i) explicitly write what was found in references 21 and 22, ii) followed by what they have found in this manuscript, and then iii) conclude what was different in the respective findings.
  5. 2 description. Sections describing Fig. 2. (figure that explains the central model) requires minor revision. Section starting from line 180 supposedly demonstrates the effects of information loss. The key concepts underlying figure 2a and 2b should be explained prior to the statements of how they are connected to thermodynamic parameters.
  6. Distributions of [what]? ‘Distribution’ is mentioned 5 times between lines 345 and 349 but it remains unclear what is meant by this term. It seems from line 344 that distributions of source nodes are being evaluated but this is not done in any of the other subsequent cases. The conclusions should also be an important section for the authors to related the term ‘source nodes’ (which is jargon) to the subject prior to the reduction applied in their approach (genes? Biochemical reactions?). This would help addressing the impact of the conclusion.
  7. Thermodynamic transitions. Line 353 comprises the closing argument for this manuscript, but reads as a free choice of words. Please specify the used terms/words: “such analysis” (does this refer to the procedure of calculate rel. FE and then rank nodes, or more?), “can” (how does this imply what the model can, and what it cannot, predict?), “inform” (meaning what?), “nodes” (meaning what?, see comment 6), “the network” (meaning which network?), “relevant” (relevant in what way?), and—importantly—“thermodynamic transitions” (is this defined anywhere in the text?).

Overall, I support the publication of this manuscript but clarification is required. Comments in 1,3, 5 in particular, are meant to clarify the story and may help convincing future readers of their approach. The points made in comment 2 and 4 are worthwhile to check. Upon checking, the conclusions may require a minor revision to specify the nature of what authors of this manuscript envision as “fundamental connection between information loss … (line 339)” and perhaps differentiate this manuscript from existing literature.

Reviewer 2 Report

This paper presents a probabilistic approach to Boolean GRNs. In this approach, each connection between two genes is treated as a communication channel and each gene state is a binary random variables. The time evolution of the network connects these random variables at consecutive times and is, in a sense, the simultaneous evolution of a statistical ensemble of networks. The stats of the network is characterized by the entropy of its nodes, and the object of study is the loss of information (or increase of entropy) across nodes and time.

The authors investigate how this loos depends on network properties, such as connectivity and ratio between positive and negative interactions. They find that increased connectivity (or accessibility score) increases the loss field significantly across the network, whereas the ratio of negative to positive interaction affects only a small number of nodes. This conclusion leads to the statement that a master regulator in the network (which is more likely in small m networks) would play a role in information transmission by decreasing loss. This is an interesting observation that relates to others in the literature on hubs and their roles in network dynamics, and this context should be discussed (for example see Harush et al., Nat Comm 2018; Rivkind et al., PLoS Comp Biol 2020). It is also noteworthy that natural GRN do have outgoing hubs and the current results can shed light on their significance; this point is only implicitly made in the paper.

For networks in transition a free energy is defined, and its distribution found also to depend sensitively on m. It is claimed that small m allows the appearance of network hubs and affects communication in the network.

The paper contains some very interesting analysis but requires clarification in several respects. It would greatly benefit from placing it in the context of other recent work on network dynamics, and from a clearer exposition of the results and conclusions.

Specific remarks:

Fig. 1: Caption does not refer to (b) and contains the unclear sentence "Each of table 2". This might be a typo or a line missing.

Fig. 2A: it is not clear how the figure matches the text, since in all parts of 2A the source node have maximal entropy. It is not clear what the arrows, colors and numbers represent exactly.

The procedure for constructing the network is not explained clearly enough. As far as I understand, the BA model is designed to crease a scale-free network. Following the pruning procedure to transform the undirected graph to a directed one, is this structure lost? If so, why start from the BA model? It would be helpful to report explicitly the distributions of outdegrees and indegrees in the network. (It is also possible to design these distributions directly, see for example the algorithm in  Kim et al., New J Phys 2012).

For the free energy associated with network transitions, the edge errors become uniformly distributed random variables. It is not clear why this modification to the model is required. Could a similar (or related) result for distributions of free energy be obtained by averaging over initial conditions? Please clarify.

The appearance of different distribution shapes as a function of m is an intriguing result, but its explanation is not clearly enough explained. In particular:

"The broader distribution suggests that the relative free energy of each replicate netork simulation is uncorrelated due to increasing interference. Hence, if a GRN has a high mean accessibility score, then the relative free energy values present in individual cells in a population are more uncorrelated with each other. "

how is m related to the correlations among replicates?

Minor comments:

L. 61: why is the analysis independent of topology? On the contrary, the results show a sensitivity to topological features such as m.

L. 63: the state of a gene is associated with a binary variable – why "bivariate"? (it is a single variable)

L. 253: why is each edge in the GRN a symmetric channel?

The term "reducible network" is mentioned several times but not defined.
